# First Detection of the West Nile Virus Koutango Lineage in Sandflies in Niger

**DOI:** 10.3390/pathogens10030257

**Published:** 2021-02-24

**Authors:** Gamou Fall, Diawo Diallo, Hadiza Soumaila, El Hadji Ndiaye, Adamou Lagare, Bacary Djilocalisse Sadio, Marie Henriette Dior Ndione, Michael Wiley, Moussa Dia, Mamadou Diop, Arame Ba, Fati Sidikou, Bienvenu Baruani Ngoy, Oumar Faye, Jean Testa, Cheikh Loucoubar, Amadou Alpha Sall, Mawlouth Diallo, Ousmane Faye

**Affiliations:** 1Pole of Virology, WHO Collaborating Center For Arbovirus and Haemorrhagic Fever Virus, Institut Pasteur, Dakar BP 220, Senegal; Bacary.SADIO@pasteur.sn (B.D.S.); Marie.NDIONE@pasteur.sn (M.H.D.N.); Moussa.DIA@pasteur.sn (M.D.); Arame.ba@pasteur.sn (A.B.); Oumar.FAYE@pasteur.sn (O.F.); Amadou.SALL@pasteur.sn (A.A.S.); Ousmane.FAYE@pasteur.sn (O.F.); 2Pole of Zoology, Medical Entomology Unit, Institut Pasteur, Dakar BP 220, Senegal; Diawo.DIALLO@pasteur.sn (D.D.); elhadji.ndiaye@pasteur.sn (E.H.N.); mawlouth.diallo@pasteur.sn (M.D.); 3Programme National de Lutte contre le Paludisme, Ministère de la Santé Publique du Niger, Niamey BP 623, Niger; hadiza_soumaila@pmivectorlink.com; 4PMI Vector Link Project, Niamey BP 11051, Niger; 5Centre de Recherche Médicale et Sanitaire, Niamey BP 10887, Niger; lagare@cermes.org (A.L.); fati@cermes.org (F.S.); jean.testa@unice.fr (J.T.); 6United States Army Medical Research Institute of Infectious Diseases, Fort Detrick, MD 21702-5011, USA; mike.wiley@unmc.edu; 7Department of Environmental, Agricultural, and Occupational Health, University of Nebraska, Omaha, NE 68198-4355, USA; 8Biostatistic, Biomathematics and Modelling Group, Institut Pasteur, Dakar BP 220, Senegal; mamadou.diop@pasteur.sn (M.D.); Cheikh.Loucoubar@pasteur.sn (C.L.); 9WHO Country Office, Niamey B.P. 10739, Niger; baruaningoyb@who.int

**Keywords:** West Nile virus, Koutango lineage, high virulence, sandflies, Niger

## Abstract

West Nile virus (WNV), belonging to the *Flaviviridae* family, causes a mosquito-borne disease and shows great genetic diversity, with at least eight different lineages. The Koutango lineage of WNV (WN-KOUTV), mostly associated with ticks and rodents in the wild, is exclusively present in Africa and shows evidence of infection in humans and high virulence in mice. In 2016, in a context of Rift Valley fever (RVF) outbreak in Niger, mosquitoes, biting midges and sandflies were collected for arbovirus isolation using cell culture, immunofluorescence and RT-PCR assays. Whole genome sequencing and in vivo replication studies using mice were later conducted on positive samples. The WN-KOUTV strain was detected in a sandfly pool. The sequence analyses and replication studies confirmed that this strain belonged to the WN-KOUTV lineage and caused 100% mortality of mice. Further studies should be done to assess what genetic traits of WN-KOUTV influence this very high virulence in mice. In addition, given the risk of WN-KOUTV to infect humans, the possibility of multiple vectors as well as birds as reservoirs of WNV, to spread the virus beyond Africa, and the increasing threats of flavivirus infections in the world, it is important to understand the potential of WN-KOUTV to emerge.

## 1. Introduction

West Nile virus (WNV) is flavivirus maintained in nature through an enzootic transmission cycle between *Culex* spp. mosquitoes including *Cx. pipiens*, *Cx. quinquefasciatus*, *Cx. neavei* and birds [1,2,3]. WN fever outbreaks occur essentially in humans and horses, considered as dead-end hosts [4]. Clinical symptoms range from asymptomatic or flu-like illness to severe neurological and meningoencephalitis syndromes [3]. WNV is one of the most widespread flaviviruses worldwide, has caused massive human and animal infections, and some fatal cases, particularly in America and Europe [5,6,7,8]. WNV has a great genetic diversity with at least eight different lineages, and of them four (lineages 1, 2, Koutango and putative new lineage 8) are present in Africa [9]. The WN lineage 1 is distributed worldwide and has been responsible for all major WN outbreaks [6,8,10,11]. The WN lineage 2 was exclusively present in Africa until 2004, when it emerged in Europe and replaced lineage 1 [12]. Migratory birds that overwintered in Africa were the most likely source of introduction of WN lineage 2 into Europe. The putative new lineage was isolated from *Cx. perfuscus* in the south-east of Senegal in 1992 and was never found associated with animals or humans [9]. The Koutango lineage of West Nile virus (WN-KOUTV) was first isolated in 1968 from the wild rodent *Tatera kempi* in Senegal [13] and in 1974 from gerbils in Somalia [14]. WN-KOUTV was initially classified as a distinct virus and later, based on phylogenetic studies, considered a WNV lineage [15,16]. WN-KOUTV is exclusively detected in Africa, and unlike other WNV lineages, it was once isolated from mosquitoes and mainly from ticks and rodents, particularly in Senegal [17]. In humans, serological evidence in Gabon [18] and a report of an accident where a Senegalese laboratory worker was symptomatically infected with WN-KOUTV [19] have been shown. Different symptoms such as two-day fever accompanied by achiness and retrobulbar headache, to erythematous eruption on the flanks, were detected [13,19]. Patients with acute febrile illness ruled out for malaria and Lassa fever in Sierra Leone were found to present neutralizing antibodies to WN-KOUTV [20]. This unpublished study shows that natural human infections with WN-KOUTV are occurring in Africa and suggests that this virus is likely the etiological agent of at least some of the fevers with unknown origin.

In animal models, intra-cerebral inoculation of the virus to new-borne mice causes death on days three to four post-infection [13], and in adult mice, WN-KOUTV showed higher virulence compared to all other WNV lineages [17,21,22]. Currently, the transmission cycle remains unclear, and the roles of ticks and mosquitoes are not yet known. Indeed, vector competence studies showed that *Cx. quinquefasciatus* and *Cx. neavei*, proven vectors for other WNV lineages, were not competent for WN-KOUTV lineage [9]. Another vector competence study showed that *Aedes aegypti* was found to carry the virus and disseminated the infection after a blood meal with high viral dose [23], suggesting that this mosquito species could probably transmit WN-KOUTV lineage to humans. It has also been shown that *Ae. aegypti* was able to vertically transmit the virus [24]. Vector competence of ticks, mostly associated with the virus in the wild, has never been tested.

In the context of Rift Valley fever (RVF) outbreak investigations in Niger, in 2016 [25], mosquitoes and sandflies were collected for arbovirus detection. The laboratory analyses revealed the presence of the WN-KOUTV lineage in a pool of sandflies. Here, we describe this first detection of WN-KOUTV in sandflies, the virus isolation, viral genome sequencing and analyses, and in vivo characterization in mice.

## 2. Results

### 2.1. Arthropod Species

A total of 10,977 hematophagous arthropods (158 mosquitoes, 10,816 sandflies and 3 biting midges) were collected and grouped into 181 pools (Table 1). Of these, sandflies were the only arthropod group collected in Intoussane, and the most abundant in Tchintabaraden (*n* = 359; 83.3%) and Tasnala (*n* = 10,428; 99.2%). *Anopheles gambiae* (58.2%) and *Cx perexiguus* (24.7%) were the most abundant among six mosquito species collected during our collection period. Sandflies were collected only by Centers for Disease Control and Prevention (CDC) light traps near herds at Intoussane, and ground pools at Tasnala, while they were found in all biotopes investigated at Tchintabaraden.

### 2.2. Virus Detection

One pool of 100 sandflies collected by CDC light trap near a ground pool at Tasnala was positive for flavivirus by immunofluorescence assay IFA and West Nile virus by real time RT-PCR. The genotyping using WNV primers and probes specific to the different lineages, as well as the genome sequencing, showed the presence of WN-KOUTV in the sample.

The minimum infection rate (MIR) was 0.01 per 1000 in the ground pool where the positive pool was collected.

### 2.3. Sequencing and Evolutionary Analyses

A genome sequence of 10,948 bp was obtained. The BLAST search showed that the sequence corresponds to WN-KOUTV and the Genbank accession number is MN057643. Phylogenetic analyses showed that the sequence of the strain isolated from sandflies belongs to the same cluster as other WN-KOUTV strains, ArD96655 (accession number KY703855.1) and Dak Ar D 5443 (accession number EU082200.2). This analysis confirmed, therefore, that the virus strain from sandflies in Niger belongs to the WN-KOUTV lineage (Figure 1).

Amino acid sequence analyses of the sandfly strain and other WNV strains showed a mean genetic distance around 0.01 within the Koutango lineage (Table 2). The sandfly strain was more closed to the rodent WN-KOUTV strain, with a genetic distance of 0.005. As with other WN-KOUTV strains, the sandfly strain also showed high genetic distances with other WNV lineages, ranging from 11 to 18% (Table 2).

Amino acid sequence alignment of the new sandfly strain and other WNV strains was performed to check for mutations that have been shown to impact WNV virulence (Figure 2). Mutations already described for WN-KOUTV strains [26,27,28] have been detected in the pre-membrane (S to M at position 72), envelope (Y to F at position 155 of the glycosylation site), and NS5 (F to S at position 653) proteins of the new sandfly WN-KOUTV strain. In addition, the rodent WN-KOUTV strain showed a specific mutation (S to P) at position 156 of the envelope protein glycosylation site. Mutations with unknown consequence (SVA to ASS), specific to all WN-KOUTV strains analyzed here, were also detected at positions 363 to 365 of the envelope protein. The new sandfly WN-KOUTV strain shared a mutation (A to T) with WNVL1 and L2 at position 366 of the envelope protein and showed specific/unique mutations compared to other WN-KOUTV strains at position 38 (M to I) of the NS2A and 177 (V to M) of the NS3 proteins (Figure 2).

### 2.4. In Vivo Characterization

Intra-cerebral inoculation of the new KOUTV strain isolated from sandflies to new-borne mice showed 100% mortality at day two post-infection, while ArD96655 showed 100% mortality at day four post-infection. In adult mice, the Koutango sandfly strain showed 100% mortality of mice at days 7 and 10 post-infection with 100 and 1000 pfu, respectively, while ArD96655 showed 100% mortality at day six post-infection with both doses (Figure 3).

In both experiments, PBS-inoculated negative control groups showed no signs of disease and stayed alive throughout the experiments.

## 3. Discussion

Our study showed, for the first time to the best of our knowledge, the isolation of WN-KOUTV from sandflies but also the detection of this particular West Nile virus lineage in Niger. Indeed, phylogenetic analyses showed that the virus strain from sandflies exhibited similar genotypic patterns to other WN-KOUTV strains already described [9,17]. WN-KOUTV was previously isolated once from mosquitoes and several times from ticks in Senegal, then this isolation in sandflies extended the spectrum of the potential WN-KOUTV vectors and highlighted once again the particular feature of this WNV lineage. The vector competence of ticks and sandflies, naturally associated with the WN-KOUTV lineage, is not proven, but in laboratory conditions it has been shown that *Ae. aegypti* can transmit the WN-KOUTV lineage, but only with a high viral dose in an artificial blood meal. This suggests that only high viremia will naturally render a vertebrate host infectious for the *Ae. aegypti* mosquito [24]. However, little is known about WN-KOUTV infections and viremia titers in vertebrate hosts. In addition, apart from rodents, the existence of other vertebrate hosts in the wild is not known. Obviously, because WN-KOUTV is a WNV lineage, birds might also play important roles in its transmission as well as propagation beyond the African continent, as proposed for lineages 1 and 2 [2,6]. All these considerations emphasize the need to better characterize this particular WNV lineage in Africa and its epidemic potential. Therefore, more studies are needed to help understand the potential of the common mosquito species *Ae. aegypti* to transmit naturally WN-KOUTV between different vertebrate hosts, and the role of birds in the transmission cycle. Vector competence studies of ticks and sandflies species are also necessary to better understand the transmission dynamics of WN-KOUTV.

Sequence analyses conducted in this study showed high genetic distances between WN-KOUTV and other WNV lineages, which confirmed that Koutango is the most distant WNV lineage [17]. The sequence alignment showed variations specific to KOUTV lineage and also between KOUTV strains. The mutations found in the envelope, pre-membrane and NS5, in all WN-KOUTV strains, could therefore explain the higher virulence of this lineage compared to other WNV lineages. In addition, although high virulence was observed for both WN-KOUTV strains in mice, differences in the survival times of newborn and adult mice were noted. Further studies with more WN-KOUTV strains are therefore needed, to better characterize the genetic variations inside the WN-KOUTV lineage and their impact on the infection in vertebrate hosts. These studies will also help to understand if, like WNV lineages 1 and 2, strains with high and low virulence exist in the WN-KOUTV lineage.

No WN-KOUTV strain was detected in the different *Culex* spp. mosquitoes collected in this study. This could be partly explained by the low number of *Culex* spp. specimens collected. However, a previous vector competence study targeting two *Culex species,* considered as the most probable WNV vectors in domestic and enzootic contexts in Senegal, also showed that they were not competent for WN-KOUTV [9]. More vector competence studies targeting different vector species and WN-KOUTV strains are needed to better characterize the role of *Culex* mosquitoes in the transmission of WN-KOUTV.

In our study, the sandfly species positive for WN-KOUTV is not known because the sandflies collected during this investigation were not identified at species level. However, seven species including *Phlebotomus roubaudi*, *Phlebotomus clydei* and *Phlebotomus orientalis* were previously collected in Niger [29,30,31] and could be targeted for vector competence studies. The very high abundance of sandflies observed around dry pools in our study is concordant with previous investigations in the Sahelian area of Senegal [32] and Mauritania [33]. The peak abundance of these sandflies was shown to occur in the Sahelian area at the beginning of the dry season, around two months after the rain pools dry up [32]. Because they are only abundant during the dry season, sandflies would probably play a role in WN-KOUTV transmission in this period, while mosquitoes and/or ticks would be the main arthropods involved during the rainy season. Many other viruses were previously detected in phlebotomine sandflies in Africa, including Chandipura virus, Saboya virus, Tete virus, and two unknown viruses in Senegal [32,34], Yellow fever virus in Uganda [35], Perinet virus in Madagascar [36], sandfly fever Sicilian and sandfly fever Naples viruses in Egypt [37], and Punique virus in Tunisia [38]. Interestingly, like WN-KOUTV, Saboya virus, another flavivirus was also detected in both sandflies and rodents in Senegal [32,39]. This emphasizes the need to better characterize and evaluate the potential roles of sandflies in arbovirus transmission cycles, because many studies are only focused on mosquitoes and ticks.

## 4. Materials and Methods

### 4.1. Collection and Processing of Arthropods

Field investigations were conducted between 20–24 October 2016, in the villages of Tchintabaraden, Intoussane and Tasnala located in the district of Tchintabaraden (15°53′53″ N; 5°48′11″ E), Tahoua Region, Niger. These villages were selected based on the presence of confirmed human RVF cases and high abortion rates in ungulates. Hematophagous arthropods (mosquitoes, sandflies and biting midges) were collected in and around households of suspected and confirmed human RVF cases, herds, and the edges of ground pools using a backpack aspirator [40], CDC light traps [41], and indoor residual spaying [42].

Arthropods collected were frozen, morphologically identified to the species level for mosquitoes using morphological keys [43,44], and family level for other arthropods, pooled by family, species, sex, and date. All arthropod pools were conserved at the laboratory in Niamey, and later aliquots were transported to Institut Pasteur de Dakar for virus testing and isolation. The minimum infection rate (MIR) was calculated to estimate the viral infection rate in arthropod populations by assuming that at least one individual of the pooled sample could be infected. The formula of MIR is as follows; MIR = number of positive pools/numbers of tested arthropods × 1000.

### 4.2. Virus Isolation

The arthropod pools were homogenized in 3 mL of L-15 medium (Gibco BRL, Grand Island, NY, USA) supplemented with 20% of fetal bovine serum and clarified by centrifugation at 1500× *g*, at 4 °C for 10 min. The supernatants were then filtered using a 1 mL syringe (Artsana, Como, Italy) and sterilized with 0.20 μm filters (Sartorius, Göttingen, Germany).

Viral isolation was conducted from the supernatants using C6-36 (Ae. albopictus) cells, and the presence of virus was detected by immunofluorescence assay (IFA) using in-house immune ascite pools specific to different flaviviruses, bunyaviruses, orbiviruses, and alphaviruses, as previously described [45].

### 4.3. RT-PCR and Titration

RNA extraction was conducted from the supernatant of the IFA-positive sample using the QiaAmp Viral RNA Extraction Kit (Qiagen, Heiden, Germany) according to the manufacturer’s instructions. The RNA samples were then screened by RT-PCR (reverse transcription-polymerase chain reaction) for flaviviruses (dengue, yellow fever, Zika and West Nile virus). The primers and probes already described elsewhere were used [46,47,48,49].

The supernatant of the IFA-positive sample (confirmed by RT-PCR) was titrated as previously described, using PS cells (Porcine Stable kidney cells, ATCC number, Manassas, VA, USA) [50].

### 4.4. Viral Sequencing and Phylogenetic Analyses

Host ribosomal RNAs were depleted prior to sequencing from extracted RNA, using specific probes from partners at the United States Army Medical Research Institute of Infectious Diseases (USAMRIID). The sequence-independent, single-primer amplification (SISPA) method was used for cDNA synthesis from depleted RNAs, and libraries were prepared using the Nextera XT library prep kit (Illumina, San Diego, CA, USA) with dual index strategy, according to the manufacturer’s instructions. Libraries were normalized and pooled with PhiX DNA as loading control, and the sequencing was performed using Miseq, Illumina, for 2 × 151 cycles. Sequencing runs were monitored in real time using the Illumina Sequencing Viewer Analyzer for cluster density, percentage of clusters passing filter, phasing/pre-phasing ratios, % base, error rates, % reads with quality score ≥30, and other parameters. The bioinformatics analyses were performed via an in-house script that implements a pipeline for pathogen discovery. De novo assembly was performed using Geneious prime v. 2019.1.3 to obtain the complete sequence. A BLAST search was then conducted to identify the assembled sequence.

Alignment and evolutionary analyses (genetic distance and phylogenetic analyses) were conducted with amino acid sequences using MEGA-X 10.2.2 software (MEGA, University of Pennsylvania, USA). Genetic distance analysis was conducted using the Poisson correction model and the phylogenetic analysis was inferred by using the maximum likelihood method and JTT (Jones, Taylor, and Thornton) matrix-based model with a bootstrap test of 1000 replicates [51,52]. The tree with the highest log likelihood is shown in Figure 1.

### 4.5. In Vivo Characterization in Mice

The IFA-positive sample was tested in newborn and adult mice in comparison with the KOUTV strain ArD96655 (accession number KY703855.1) isolated from ticks.

Ten new-borne Swiss mice (1–2 days old) were inoculated with 1000 pfu of IFA positive sample, ArD96655 and PBS alone, by an intra-cerebral route and were monitored daily for 21 days.

Five- to six-week-old Swiss mice were also challenged by intraperitoneal route with 100 pfu and 1000 pfu, to analyze the virulence of the IFA-positive sample. The WN-KOUTV strain ArD96655 and PBS were also tested as positive and negative controls, respectively. Eight mice were tested for each dose. Survival curves were generated using GraphPad Prism 9.0.0 (121) software (GraphPad, San Diego, CA, USA).

## 5. Conclusions

In this study, we have shown the circulation of WN-KOUTV in Niger and its detection in sandflies for the first time. These results extended the number of countries in Africa where this virus is reported, but also the spectrum of potential vectors. The very high virulence in mice [17,21,22], the possibility of multiple vectors, the risk of KOUTV to infect humans, and the increasing threats of flavivirus infections in the world, should contribute to better consideration of WNV-KOUTV as an important emerging pathogen. In this regard, vector competence studies, vertebrate hosts including birds, and viral genetic diversity characterizations are ongoing and will provide new insights on WN-KOUTV transmission, virulence and possibility to diffuse beyond Africa. Seroprevalence studies would also be important in Niger, particularly in Tahoua Region where the virus was isolated, and in Senegal where WN-KOUTV was isolated several times, to assess the potential circulation of KOUTV in humans.

## Figures and Tables

**Figure 1 pathogens-10-00257-f001:**
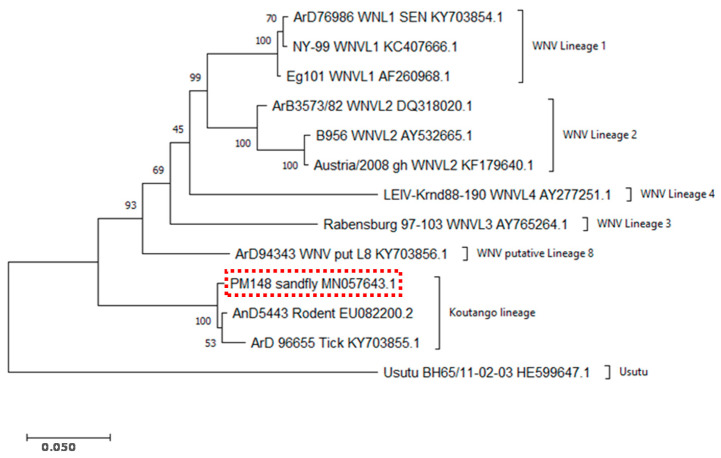
Phylogenetic analyses using MEGA software and the maximum likelihood method. Phylogenetic analyses were conducted with sequences of the sandfly strain, other Koutango strains isolated in Senegal from ticks and rodents, and other WNV lineage strains. The accession numbers and names of the strains are mentioned. The accession number of the sandfly strain genomic sequence, PM148 from Niger, is MN057643. WNVL1: lineage 1, WNVL2: lineage 2, WNVL3: lineage 3, WNVL4: lineage 4 and WNV Put L8: putative lineage 8.

**Figure 2 pathogens-10-00257-f002:**
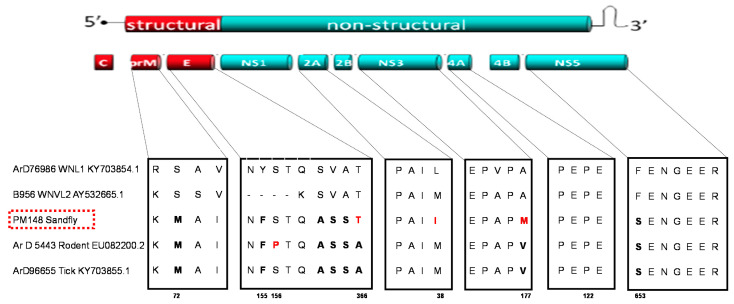
Genetic diversity of WNV lineages 1, 2 and Koutango. Alignment was conducted with sequences of the sandfly strain, other Koutango strains isolated in Senegal from ticks and rodents, and other WNV lineages 1 and 2 strains. The genomic structure of West Nile virus is shown, and the different genes are labeled. Alignments of motifs with unknown consequence and known virulence motifs are shown. Mutations specific to all Koutango strains are in bold and mutations specific to one Koutango strain are in red.

**Figure 3 pathogens-10-00257-f003:**
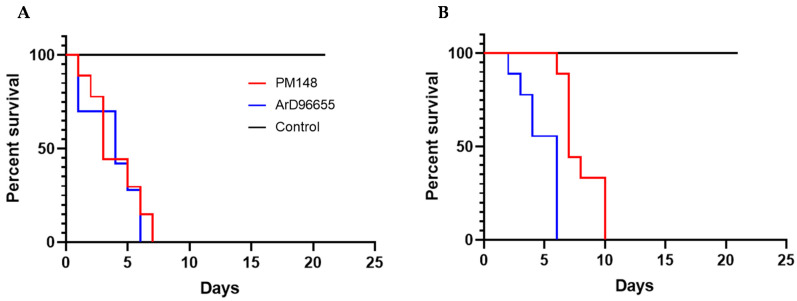
Survival curves of 5- to 6-week-old mice following intraperitoneal infection with (**A**) 100, and (**B**) 1000 pfu. Eight mice were tested for each dose. A group of mice with an injection of PBS was used as control. Mice were monitored daily for 21 days.

**Table 1 pathogens-10-00257-t001:** List of arthropods collected in the field from three districts of Niger, 20–24 October 2016.

Species	Tchintabaraden	Intoussane	Tasnala	Total
*Anopheles gambiae*	40		52	92
*Anopheles rufipes*	1		3	4
*Cx. antennatus*	1			1
*Cx. ethiopicus*	1		3	4
*Cx. quinquefasciatus*	14		4	18
*Cx. perexiguus*	14		25	39
Total mosquitoes	71		87	158
Biting midges	1		2	3
Sandflies	359	29	10,428	10,816
Total arthropods	431	29	10,517	10,977

**Table 2 pathogens-10-00257-t002:** Genetic distance analysis conducted using MEGA software with the Poisson correction model.

Strains	1	2	3	4	5	6	7	8	9	10	11	12
1.PM148_sandfly_MN057643.1_Koutango_lineage												
2.AnD5443_Rodent_EU082200.2_Koutango_lineage	0.00584											
3.ArD_96655_Tick_KY703855.1_Koutango_lineage	0.02121	0.00848										
4.ArD76986_SEN_KY703854.1_WNV_Lineage_1	0.11371	0.11469	0.11764									
5.Eg101_AF260968.1_WNV_Lineage_1	0.13990	0.11208	0.14274	0.00584								
6.NY-99_KC407666.1_WNV_Lineage_1	0.14174	0.11404	0.14398	0.00467	0.00769							
7.ArB3573/82_DQ318020.1_WNV_Lineage_2	0.13645	0.11371	0.14039	0.06095	0.07205	0.07557						
8.Austria/2008_gh_KF179640.1_WNV_Lineage_2	0.13084	0.11310	0.12682	0.06004	0.08429	0.08696	0.03494					
9.B956_AY532665.1_WNV_Lineage_2	0.13294	0.11353	0.12892	0.06196	0.08603	0.08930	0.03663	0.00634				
10.ArD94343_KY703856.1_WNV_putative_Lineage_8	0.12553	0.12752	0.12917	0.09528	0.09432	0.09368	0.09176	0.09147	0.09156			
11.LEIV-Krnd88-190_AY277251.1_WNV_Lineage_4	0.18058	0.15722	0.18384	0.12070	0.13969	0.14094	0.13333	0.13487	0.13640	0.14568		
12.Rabensburg_97-103_AY765264.1_WNV_Lineage_3	0.16703	0.14386	0.17689	0.10107	0.12331	0.12359	0.11968	0.12823	0.13222	0.12224	0.16220	

In grey, genetic distances within WN-KOUTV lineage. Red rectangle, genetic distances between sandfly strain and with other WNV lineages.

## Data Availability

All data are included in the manuscript. The virus sequence generated during the current study is available at Genbank (accession number: MN057643).

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
