# Peer review of "First Detection of the West Nile Virus Koutango Lineage in Sandflies in Niger"

_pathogens, 2021, doi:10.3390/pathogens10030257_

Round 1
Reviewer 1 Report
Dear authors,
Please find the review suggestions bellow.
General:
As long as this paper describes the detection of WN-KOUTV, all the details regarding RVF are redundant, please remove. The last paragraph in the Discussions section (L224-227), which is supported neither by the aim/hypothesis nor the results given in the study
Title and further on
As Gamou et al (2017) show (https://journals.plos.org/plosntds/article?id=10.1371/journal.pntd.0006078) ‘Koutango virus (lineage 7) was initially classified as a different virus, but is now a distinct lineage of WN virus’. It is clear that it is particular so, specifying this sounds redundant. Also naming this lineage a virus and then specifying that it is a lineage creates a sense of confusion. Specifying the year in the title is not often seen but if you consider this being of high relevance, than keep it. Taking these into account a title suggestion would be ‘First detection of the West Nile virus Koutango lineage in sandflies in Niger’
Abstract
L18: although grammatically may be correct (‘-borne describing the method or means by which something is carried or moved), the ‘mosquito-borne virus’ expression is not usual (compared to the widely used ‘mosquito-borne disease/illness/infection’), probably considering that it is not the most important defining trait of the pathogen (but it is a crucial one in case of the disease produced), please rephrase here and later on
L20: consider to be more specific regarding the new putative lineage
L22-24: is the motive and result of insect collection that important that it needs to be stated in the Abstract, as long as it is not truly relevant for the subject of the manuscript? Also, the sentence is not clear enough in L24, whose geno-and phenotypic characterization and what positive sample if the insects were negative for RVF virus? Please rephrase
L26: please rephrase ‘sandfly strain clustered with other WN-KOUTV strains’ as it is not clear
L29: please be more specific regarding the ‘probable multiple vectors’/change to ‘possibility of multiple vectors’ and if you use the latter then rephrase the sentence so as ‘possibility’ won’t be repeated (can be changed to ‘risk’ regarding human infections)
L31: here and especially later in the discussions: please address the other epidemiological factors that should be involved too to support (or drop) the affirmation about the WN-KOUTV having the potential to emerge as an important global threat
L29: consider to explain more in depth the ‘possibility of WN-KOUTV to infect humans’
L31: please consider the cardinal importance of reliable and evidence based scientifical data sharing (and also possible impact on the readers) when using strong expressions such as ‘important global threat’
Introduction
L36: please see comment for L18
L44: please specify the species you concern here
L44: change ‘up to’ to ‘until’
L46: please insert ‘WN’ before ‘lineage 2’
L47: insert ‘The’ before ‘Koutango lineage’
L62-63: rephrase ‘higher virulence among all WNV lineages’ either by keeping ‘higher’ and stating compared to what or changing to ‘highest’ if this was the case; ‘within’ may be better to be used instead of ‘among’ in the latter case
L64: please consider to rephrase the ‘-borne’ expression
L67: consider to rephrase ‘competent with high viral dose’ (able to harbor/found to carry)
Results
L78-92: this part doesn’t present results of the study but materials and methods, please move to the appropriate section
L94: first sentence is redundant, please delete (and delete ‘however’ too)
L99: please spell out the abbreviations at their first usage (MIR)
L101: first sentence belongs to the materials and methods, please move there
L116-117: change ‘The sandfly strain is more closed‘ to ‘The sandfly strain was more close’
L119: change ‘lineage’ to ‘lineages’
L128: ‘shared a mutation’ or ‘shared mutations’
Discussion
L162: change ‘spectra’ to ‘spectrum’ (singular)
L163: please clarify more what do you mean by ‘the particular feature of this WNV lineage’
L166-167: this sentence is redundant, repeating the information already stated in the previous one (with the same reference too). Please delete or combine into a single sentence with the previous one (without redundancy)
L169: ‘interestingly’ is not the appropriate word here, please change (obviously?)
L173: see the comment for L31
L163-180: please shorten and condense this part avoiding different phrasing of the same information
L181: change ‘sequences analyses’ to ‘the sequence analyses’
L182: lineages and strains are not the same and please don’t use both terms together. Did you mean strains of/from other lineages?
L191: delete the word ‘indeed’
L191-193: redundant sentence. Please delete/rephrase
L194: ‘phenotypic’ is not the appropriate word here, please change
L194-197: redundant sentence, this was said already before, please correct
L202: please don’t use together ‘probably’ and ‘suggests’. Suggesting already implies incertitude
L205: the need for characterization of the main WN-KOUTV vectors was already stated, please don’t repeat
L206: change ‘collected sandflies’ to ‘the sandflies collected’
L214: remove the comma after ‘because’
L215: insert a comma between ‘period’ and ‘while’
L215: use ‘would’ instead of ‘will’ (parallelism on the sentence)
L216-221: please be aware that the relevance of this paragraph is minimal for the study performed. Consider to change.
L224-227: this affirmation is not supported by any of the results of this study, please remove (It wasn’t either hypothesized nor part of the aim of the study, also)
Materials and Methods
The description of the statistical methods is missing, please complete
L241: use either ‘arthropod pools’ or ‘arthropods’
L241-242: consider to change to either ‘at the laboratory in Niamey’ or ‘in laboratory conditions at Niamey’
L243: delete ‘attempts’
L244: specimens instead of specimen
L246: Insert ‘the’ before ‘arthropod
L247: change ‘clarify’ to ‘clarified’
L250: reword ‘attempted’
L255: change to ‘The supernatant of/from the IFA positive samples was used for RNA extraction with the help of the…’
L260: ‘the IFA positive samples’
L265: insert ‘the’ before ‘United’
L266: change ‘independent’ to ‘independent’
L270: insert ‘the’ before ‘sequencing’
L284: the location of the mice producing farm is not relevant for the study. Please remove.
Conclusions
L299: change ‘spectra’ to ‘spectrum’
L299-303: see the comments for the same sentence in the Abstract
L306: if you are sure that these studies will be performed, please state this explicitly and use ‘will’. If not, please use ‘would’. The same applies for the future tense expression in L307
Author Response
I would like to thank the reviewer 1. Below the answers:
Response to Reviewer 1 Comments
General
Point 1: As long as this paper describes the detection of WN-KOUTV, all the details regarding RVF are redundant, please remove. The last paragraph in the Discussions section (L224-227), which is supported neither by the aim/hypothesis nor the results given in the study
Response 1: All the details regarding RVF, including the last paragraph of the discussion section were removed. See the revised manuscript.
Title and abstract
Point 2: As Gamou et al (2017) show (https://journals.plos.org/plosntds/article?id=10.1371/journal.pntd.0006078) ‘Koutango virus (lineage 7) was initially classified as a different virus, but is now a distinct lineage of WN virus’. It is clear that it is particular so, specifying this sounds redundant. Also naming this lineage a virus and then specifying that it is a lineage creates a sense of confusion. Specifying the year in the title is not often seen but if you consider this being of high relevance, than keep it. Taking these into account a title suggestion would be ‘First detection of the West Nile virus Koutango lineage in sandflies in Niger’
Response 2: The title was changed to ‘First detection of the West Nile virus Koutango lineage in sandflies in Niger’. See L2-3 of the revised manuscript.
Point 3: L18: although grammatically may be correct (‘-borne describing the method or means by which something is carried or moved), the ‘mosquito-borne virus’ expression is not usual (compared to the widely used ‘mosquito-borne disease/illness/infection’), probably considering that it is not the most important defining trait of the pathogen (but it is a crucial one in case of the disease produced), please rephrase here and later on
Response 3: This part was rephrased and we used mosquito-borne disease instead of mosquito-borne virus. See L19-20 of the revised manuscript.
Point 4: L20: consider to be more specific regarding the new putative lineage
Response 4: The putative new lineage of WNV was detected in 1992 in the South-East of Senegal, and already sequenced and characterized in Culex mosquitoes and mice (Fall et al., 2014; Fall et al., 2017).
This part was deleted in the abstract and the following sentence was added in the introduction “The putative new lineage was isolated from Culex perfuscus in the South-East of Senegal in 1992 and was never found associated with animals or humans”. See L49-51 of revised manuscript.
Point 5: L22-24: is the motive and result of insect collection that important that it needs to be stated in the Abstract, as long as it is not truly relevant for the subject of the manuscript? Also, the sentence is not clear enough in L24, whose geno-and phenotypic characterization and what positive sample if the insects were negative for RVF virus? Please rephrase
Response 5: This part was rephrased. We just kept the context in which the samples were collected but all results on Rift valley fever are removed.
Genotypic (Whole genome sequencing and sequences analyses) and phenotypic (replication studies in mice) were conducted. Corrected in the abstract and the introduction. See L25-26 of revised manuscript
Point 6: L26: please rephrase ‘sandfly strain clustered with other WN-KOUTV strains’ as it is not clear
Response 6: Rephrased as suggested: “analyses confirmed that this strain belongs to the WN-KOUTV lineage”. See L27-28 of revised manuscript.
Point 7: L29: please be more specific regarding the ‘probable multiple vectors’/change to ‘possibility of multiple vectors’ and if you use the latter then rephrase the sentence so as ‘possibility’ won’t be repeated (can be changed to ‘risk’ regarding human infections)
Response 7: This part was rephrased as follows: given the risk of WN-KOUTV to infect humans, the possibility of multiple vectors etc… See L30-33 of the revised manuscript.
Point 8: L31: here and especially later in the discussions: please address the other epidemiological factors that should be involved too to support (or drop) the affirmation about the WN-KOUTV having the potential to emerge as an important global threat
Response 8: We added ‘the possibility of birds as reservoirs of WNV, to spread the virus beyond Africa’ as another important epidemiological factor. See L31 and L169-171 of the revised manuscript.
-
Point 9: L29: consider to explain more in depth the ‘possibility of WN-KOUTV to infect humans’
Response 9: We consider that the detection of antibodies neutralizing WN-KOUTV in humans and the accidental infection in laboratory showed that human infections with this lineage are possible that’s the reason why we talk about “possibility of human infections”. We now change to risk of WN-KOUTV to infect humans. See L30 and 289 of the revised manuscript.
Point 10: L31: please consider the cardinal importance of reliable and evidence based scientifical data sharing (and also possible impact on the readers) when using strong expressions such as ‘important global threat’
Response 10: We agree with you and this part was rephrased as follows: “it is important to understand the potential of WN-KOUTV to emerge”, see L32-33 and to better consider this WN-KOUTV as an important emerging pathogen, see L288-291 of revised manuscript.
Introduction
Point 11: L36: please see comment for L18
Response 11: This part was removed now.
Point 12 : L44: please specify the species you concern here
Response 12 : Many species of Culex have already been shown to transmit WNV. I added Culex quinquefasciatus, Culex pipiens, Culex neavei as examples. See L38-39 of revised manuscript.
Point 13 : L44: change ‘up to’ to ‘until’
Response 13 : « up to » was changed to « until ». See L47 of revised manuscript.
Point 14 : L46: please insert ‘WN’ before ‘lineage 2’
Response 14 : « WN » was inserted, see L49 of revised manuscript.
Point 15 : L47: insert ‘The’ before ‘Koutango lineage’
Response 15 : « The » was inserted before Koutango lineage, see L51 of revised manuscript.
Point 16 : L62-63: rephrase ‘higher virulence among all WNV lineages’ either by keeping ‘higher’ and stating compared to what or changing to ‘highest’ if this was the case; ‘within’ may be better to be used instead of ‘among’ in the latter case
Response 16 : We rephrased as follows : WN-KOUTV showed higher virulence compared to all other WNV lineages. See L66-67 of the revised manuscript.
Point 17 : L64: please consider to rephrase the ‘-borne’ expression
Response 17 : We removed the –borne- expressions and rephrased as follows : « Currently, the transmission cycle remains unclear and the roles of ticks and mosquitoes are not known yet. » See L68 of the revised manuscript.
Point 18 : L67: consider to rephrase ‘competent with high viral dose’ (able to harbor/found to carry)
Response 18 : This sentence was rephrased as follows « While another vector competence study showed that Aedes aegypti was found to carry the virus and disseminate the infection after a blood meal with high viral dose. It has also been shown that Aedes aegypti was able to transmit vertically the virus». See L70-72 of the revised manuscript.
Results
Point 19 : L78-92: this part doesn’t present results of the study but materials and methods, please move to the appropriate section
Response 19: this part was deleted as it was already in the materials and methods
Point 20 : L94: first sentence is redundant, please delete (and delete ‘however’ too)
Response 20: The sentence and ‘however ‘ were deleted
Point 21 : L99: please spell out the abbreviations at their first usage (MIR)
Response 21: Done, see L98 of the revised manuscript.
Point 22 : L101: first sentence belongs to the materials and methods, please move there
Response 22: this part was removed
Point 23 : L116-117: change ‘The sandfly strain is more closed‘ to ‘The sandfly strain was more close’,
Response 23 : Done, the sentence was corrected, see L117 of the revised manuscript.
Point 24 : L119: change ‘lineage’ to ‘lineages’
Response 24 : corrected, see L119 of the revised manuscript.
Point 25 : L128: ‘shared a mutation’ or ‘shared mutations’
Response 25 : Corrected to « shared a mutation » see L128 of the revised manuscript.
Discussion
Point 26 : L162: change ‘spectra’ to ‘spectrum’ (singular)
Response 26 : spectrum was corrected. See L161 of the revised manuscript.
Point 27 : L163: please clarify more what do you mean by ‘the particular feature of this WNV lineage’
Response 27 : This lineage has been isolated from 3 different arthropods (mosquitoes, ticks and sandflies) and mainly from ticks. This feature is particular compared to other WNV lineages mostly isolated from mosquitoes.
Point 28 : L166-167: this sentence is redundant, repeating the information already stated in the previous one (with the same reference too). Please delete or combine into a single sentence with the previous one (without redundancy)
Response 28 : Here, I am talking in the first sentence about the experimental infections showing that with high viral dose in the blood meal, Ae aegypti is competent and able to transmit WN-KOUTV. In the second sentence, the authors made hypothesis regarding real life and possible infection of Ae aegypti from infectious vertebrates.
I rephrased to make it more clear
« it has been shown that Ae. aegypti can in laboratory conditions transmit WN-KOUTV lineage but only with high viral dose in the artificial blood meal. This suggests that only high viremia will naturally render a vertebrate host infectious for Ae. aegypti mosquito ». See L164-167 of the revised manuscript.
Point 29 : L169: ‘interestingly’ is not the appropriate word here, please change (obviously?)
Response 29 : Interestingly was changed to obviously. See L169 of the revised manuscript.
Point 30 : L173: see the comment for L31
Response 30 : « at global level » was removed.
Point 31 : L163-180: please shorten and condense this part avoiding different phrasing of the same information
Response 31 : This part is shorten and redundant parts are deleted. See L163-177 of the revised manuscript.
Point 32 : L181: change ‘sequences analyses’ to ‘the sequence analyses’
Response 32 : Sequence was corrected. See L178 of the revised manuscript.
Point 33 : L182: lineages and strains are not the same and please don’t use both terms together. Did you mean strains of/from other lineages?
Response 33 : Corrected to «other lineages ». See L179 of the revised manuscript.
Point 34 : L191: delete the word ‘indeed’
Response 34 : Indeed was deleted
Point 35 : L191-193: redundant sentence. Please delete/rephrase
Response 35 : the sentence was deleted
Point 36 : L194: ‘phenotypic’ is not the appropriate word here, please change
Response 36: This part was rephrased according to another reviewer suggestion and phenotypic was removed. See L185-87 of the revised manuscript.
Point 37 : L194-197: redundant sentence, this was said already before, please correct
Response 37 : The sentence was removed.
Point 38 : L202: please don’t use together ‘probably’ and ‘suggests’. Suggesting already implies incertitude
Response 38 : This part was rephrased according to another reviewer suggestion and ‘probably suggest’ was removed. See L190-96 of the revised manuscript.
Point 39 : L205: the need for characterization of the main WN-KOUTV vectors was already stated, please don’t repeat
Response 39 : We focused on Culex mosquitoes in this part and rephrased as follows : « More vector competence studies targeting different vector species and WN-KOUTV strains are needed to characterize the role of Culex mosquitoes in the transmission of WN-KOUTV ». See L194-96 of the revised manuscript.
Point 40 : L206: change ‘collected sandflies’ to ‘the sandflies collected’
Response 40 : Done. See L197-98 of the revised manuscript.
Point 41 : L214: remove the comma after ‘because’
Response 41 : Done, See L204 of the revised manuscript.
Point 42 : L215: insert a comma between ‘period’ and ‘while’
Response 42: Done. See L206 of the revised manuscript.
Point 43 : L215: use ‘would’ instead of ‘will’ (parallelism on the sentence)
Response 43 : Done. See L206 of the revised manuscript.
Point 44 : L216-221: please be aware that the relevance of this paragraph is minimal for the study performed. Consider to change.
Response 44 : Here, we would like to emphasizes the need to better characterize and evaluate the potential roles of sandflies in the transmission of arboviruses, as usually experimental infections in laboratory mainly focused on mosquitoes and ticks. We reworded to clarify. See L207-215 See of the revised manuscript.
Point 45 : L224-227 : this affirmation is not supported by any of the results of this study, please remove (It wasn’t either hypothesized nor part of the aim of the study, also)
Response 45 : This part on RVF and was removed in the revised version.
Materials and Methods
Point 46 : The description of the statistical methods is missing, please complete
Response 46 : The description is done now : «The minimum infection rate (MIR) was calculated to estimate the viral infection rate in arthropod populations by assuming that at least one individual of the pooled sample could be infected. The formula of MIR is as follows; MIR= number of positive pools/numbers of tested arthropods× 1,000 ». See L230-233 of the revised manuscript.
Point 47 : L241: use either ‘arthropod pools’ or ‘arthropods’
Response 47 : we used «arthropods ». See L226 of the revised manuscript.
Point 48 : L241-242: consider to change to either ‘at the laboratory in Niamey’ or ‘in laboratory conditions at Niamey’
Response 48 : We used « at the laboratory in Niamey ». See L228-29 of the revised manuscript.
Point 49 : L243: delete ‘attempts’
Response 49 : « attemps » was deleted.
Point 50 : L244: specimens instead of specimen
Response 50 : This part was finally deleted
Point 51 : L246: Insert ‘the’ before ‘arthropod
Response 51 : « The » was inserted. See L235 of the revised manuscript.
Point 52 : L247: change ‘clarify’ to ‘clarified’
Response 52 : corrected, see L236 of the revised manuscript.
Point 53 : L250: reword ‘attempted’
Response 52 : we used « done ». See L239 of the revised manuscript.
Point 54 : L255: change to ‘The supernatant of/from the IFA positive samples was used for RNA extraction with the help of the…’
Response 54 : we rephrased as follows : RNA extraction was conducted from the supernatant of the IFA positive sample using the QiaAmp Viral RNA Extraction Kit ». See L244 of the revised manuscript.
Point 55 : L260: ‘the IFA positive samples’
Response 55 : There was one IFA positive sample, corrected. See L249 of the revised manuscript.
Point 56 : L265: insert ‘the’ before ‘United’
Response 56 : « The » was inserted. See L254 of the revised manuscript.
Point 57 : L266: change ‘independent’ to ‘independent’
Response 57 : « independent » was corrected ; L255 of the revised manuscript.
Point 58 : L270: insert ‘the’ before ‘sequencing’
Response 58 : « The » was inserted ; L259 of the revised manuscript.
Point 59 : L284 : the location of the mice producing ar mis not relevant for the study. Please remove.
Response 59 : This part was removed.
Conclusions
Point 60 : L299: change ‘spectra’ to ‘spectrum’
Response 60 : corrected, see L287 of the revised manuscript.
Point 61 : L299-303: see the comments for the same sentence in the Abstract
Response 61 : We rephrased as follows « Given the very high virulence in mice, the possibility of multiple vectors, as well as birds to be involved as reservoirs, the risk of KOUTV to infect humans and the increasing threats of flavivirus infections in the world, it is important to understand the potential of KOUTV to emerge as a significant pathogen ». See L287-291 of the revised manuscript.
Point 62 : L306: if you are sure that these studies will be performed, please state this explicitly and use ‘will’. If not, please use ‘would’. The same applies for the future tense expression in L307
Response 62: Some of these studies are already ongoing. We rephrased as follows: « In this regard, vector competence studies, vertebrate hosts including birds and viral genetic diversity characterizations are ongoing and will give new insights on WN-KOUTV transmission, virulence and possibility to diffuse beyond Africa. Seroprevalence studies also would be important in Niger, particularly in Tahoua region where the virus was isolated, and in Senegal where WN-KOUTV was isolated several times, to assess the potential circulation of KOUTV in humans. See L291 and 294 of the revised manuscript.
Reviewer 2 Report
The research work is interesting, drawing public attention to new WNV lineages in extended geographical areas, with potentially higher pathogenicity for human dead end hosts. The work is a premier by identifying for the first time the presence of WN-KOUTV in Niger in 2016. It is unclear why the authors, given the emerging importance of the WN-KOUTV, did not report its presence before.
There are some rather unclear statements, ie, MIR calculation ("the minimum infection rate (MIR), is calculated: ([number of positive pools / total specimens tested] x 1000)), how many Swiss mice were used to evaluate the the virulence of the IFA positive sample.
Similarly, the statements "The mutations specific to the sandfly 193 strain, found in our study, might explain these phenotypic differences in mice" in connection with differences in the survival times of different mice categories and also "These mutations as 187 well as the unknown mutations found in all WN-KOUTV strains, could therefore explain 188 the higher virulence of this lineage compared to other WNV lineages." sound rather speculative.
The association of WN-KOUTV strains mainly associated with sandflies and RVF outbreak should be more explicitly explained. The authors should mention under which ethical provisions they carried out the in vivo experiments.
Author Response
I would like to thank the reviewer 2. Below my answers:
Point 1 : The research work is interesting, drawing public attention to new WNV lineages in extended geographical areas, with potentially higher pathogenicity for human dead end hosts. The work is a premier by identifying for the first time the presence of WN-KOUTV in Niger in 2016. It is unclear why the authors, given the emerging importance of the WN-KOUTV, did not report its presence before.
Response 1 : The principal investigators of this study are from Senegal and were in Niger for the first time as experts for the World Health Organization, to investigate an outbreak of Rift Valley Fever in this country. As usual, primary screening was done in Niger for Rift Valley fever virus (RVFV) detection. Since all samples were negative for RVFV, a screening using cell culture, immunofluorescence and RT-PCR assays was later done in Senegal for detection of any other arbovirus in the samples. This technique is able to identify 70 arboviruses circulating in Africa. It is in this context that the WN-KOUTV was isolated in this country for the first time. The presence of this virus in Senegal was already reported in ticks and wild rodents. See “1. Coz et al. Cah ORSTOM Ser Ent Med Parasitol 1975, 13(2), 57–62.”2. Fall et al. PLoS Negl. Trop. Dis. 2017, 11, e0006078, doi:10.1371/journal.pntd.0006078.
Point 2 : There are some rather unclear statements, ie, MIR calculation ("the minimum infection rate (MIR), is calculated: ([number of positive pools / total specimens tested] x 1000)), how many Swiss mice were used to evaluate the the virulence of the IFA positive sample.
Response 2
We reworded to clarify. See the revised manuscript:
- “The minimum infection rate (MIR) was calculated to estimate the viral infection rate in arthropod populations by assuming that at least one individual of the pooled sample could be infected. The formula of MIR is as follows; MIR= number of positive pools/numbers of tested arthropods× 1,000. See L230-233 of the revised manuscript.
- As indicated in the Materials and methods section, 10 newborn mice and 8 aduld mice were used for each condition regarding the in vivo L276 and 281 of the revised manuscript.
Point 3 : Similarly, the statements "The mutations specific to the sandfly 193 strain, found in our study, might explain these phenotypic differences in mice" in connection with differences in the survival times of different mice categories and also "These mutations as 187 well as the unknown mutations found in all WN-KOUTV strains, could therefore explain 188 the higher virulence of this lineage compared to other WNV lineages." sound rather speculative.
Response 3 : The aim here was to see if phenotypic variations in animals (ie. survival times of mice) could be linked to genetic variations between the different viral strains. We agree that our statements are speculative and know that reverse genetic studies are needed to prove this link. Therefore, we rephrased this part of the document as follows: ”The mutations found in the envelope, pre-membrane and NS5, in all WN-KOUTV strains could therefore explain the higher virulence of this lineage compared to other WNV lineages. In addition, although high virulence was observed for both WN-KOUTV strains in mice, differences in the survival times of newborn and adult mice were noted. Further studies with more WN-KOUTV strains are therefore needed, to better characterize the genetic variations inside the WN-KOUTV lineage and their impact on host infections” See L181-187 of the revised manuscript.
Point 4 : The association of WN-KOUTV strains mainly associated with sandflies and RVF outbreak should be more explicitly explained. The authors should mention under which ethical provisions they carried out the in vivo experiments.
Response 4
- The Rift Valley fever outbreak occured in Niger between September and December 2016. Entomological investigations conducted in the affected region in October 2016 did unfortunately not allow us to identify the RVFV vectors. However, the screening of the collected arthropods, showed presence of Koutango virus during the RVF outbreak. This feature of detection of another arbovirus during a RVF outbreak occurred several times in West Africa. We therefore think that this is important to consider. However, following the reviewer 1 suggestion, all the details regarding RVF, including the last paragraph of the discussion section were removed. See the revised manuscript.
- Regarding ethics, these experiments are done through viro-entomological surveillanceactivities, conducted at the WHO collaborating center for arboviruses and hemorrhagic fever viruses at the Institut Pasteur de Dakar. In addition, as indicated in the Institutional Review Board Statement section, there is no National or Institutional Ethics Committee for animals in Senegal where protocols should be submitted to get approval. Finally, we always make sure in our animal facilities that any study involving animals respect the World Organization for Animal Health regulations (https://www.oie.int/fileadmin/Home/fr/Health_standards/tahc/current/chapitre_aw_research_education.pdf). Knowing the crucial role of ethical approval for animal studies, an ad hoc committee is recently created at the Vetenarian Scool at the University, where all our next projects involving animals will be submitted for approval. See L306-310 of the revised manuscript.
Reviewer 3 Report
I enjoyed reading this manuscript.
The most blatant error I found was the omission of ethical data and statements. This study performed viral infections in mice. Please provide your approval data and follow the journal guidelines for including animal data.
There were minor grammatical errors throughout that should be corrected. For instance, Line 26 should read "the sand-fly". Line 50 should read "considered a WNV lineage."
The authors state that WN-KOUTV was not found in mosquitos. The data show this however, the authors need to emphasize that their sample size was very small. The sample size was too small to make the conclusion that culex mosquitoes are not a vector.
Author Response
Response to Reviewer Comments
Point 1: The most blatant error I found was the omission of ethical data and statements. This study performed viral infections in mice. Please provide your approval data and follow the journal guidelines for including animal data
Response 1: Regarding ethics, these experiments are done through viro-entomological surveillance activities conducted at the WHO collaborating center for arboviruses and hemorrhagic fever viruses at the Institut Pasteur de Dakar. In addition, as indicated in the Institutional Review Board Statement section, there is no National or institutional Ethics Committee for animals in Senegal where protocols should be submitted to get approval. Finally, we always make sure in our animal facilities that any study involving animals respect the World Organization for Animal Health regulations (https://www.oie.int/fileadmin/Home/fr/Health_standards/tahc/current/chapitre_aw_research_education.pdf). Knowing the crucial role of ethical approval for animal studies, an ad hoc committee is recently created at the Vetenarian Scool at the University, where all our next projects involving animals will be submitted for approval. See L306-310 of the revised manuscript.
Point 2: There were minor grammatical errors throughout that should be corrected. For instance, Line 26 should read "the sand-fly". Line 50 should read "considered a WNV lineage."
Response 2: We corrected grammatical errors.
Sand-fly was finally removed because this part was rephrased
“Considered a WNV lineage” was also corrected. See L54 of the revised manuscript.
Point 3: The authors state that WN-KOUTV was not found in mosquitos. The data show this however, the authors need to emphasize that their sample size was very small. The sample size was too small to make the conclusion that culex mosquitoes are not a vector.
Response 3: we added a sentence to emphasize the low sample size of Culex mosquitoes and indicated that despite previous results suggesting that two Culex species were not competent to WN-KOUTV, more studies are needed to conclude on the main vectors of this virus. See L191-196 of the revised manuscript.